# SINGLE DOMAIN GENERALIZATION FOR RARE EVENT DETECTION IN MEDICAL IMAGING

## ABSTRACT

Single Domain Generalization (SDG) addresses the challenge of training a model on a single domain to ensure it generalizes well to unseen target domains. Although extensively studied in image classification, there is a lack of prior work on SDG for rare event or image classification in imbalanced dataset. In the medical diagnosis and disease detection domain, where data is often limited and events of interest are rare, deep learning (DL) models frequently exhibit suboptimal performance, leading to poor generalization across datasets. In multi-center studies, disparate data sources, differences in scanners and imaging protocols introduce domain shifts that exacerbate variability in rare event characteristics. This paper addresses this challenge by first leveraging a pre-trained large vision model to rank classes based on their similarity to the rare event class, allowing focused handling of the most similar class, and then integrates domain-invariant knowledge on rare event with DL to accurately classify the rare event class. By carefully incorporating expert knowledge with data-driven DL, our technique effectively regularizes the model, enhancing robustness and performance even with limited data availability. We present a case study on seizure onset zone detection using fMRI data, demonstrating that our approach significantly outperforms state-of-the-art vision transformers, large vision models, and knowledge-based systems, achieving an average F1 score of 90.2% while maintaining an overall F1 score of 85.0% across multi-center datasets.

## 1 INTRODUCTION

Out-of-distribution (OOD) generalizability refers to the model's ability to perform well on data that differs from the data used during training Stolte et al. (2023). This concept is particularly crucial in the field of artificial intelligence (AI) for medicine, where the aim is to accurately diagnose new patient cases across centers. In medical imaging, there are various factors such as age, gender, the type of imaging scanner, sedation levels, and coexisting conditions that cause differences in how a disease appears in images Bera & Biswas (2023)Chen et al. (2023a)Chen et al. (2023b)Chen et al. (2023c)Hu et al. (2023)Parker et al. (2023). These differences, known as intra-class variabilityChe et al. (2023)Liu et al. (2023), can challenge DL because there might be limited data for certain conditions, affecting the model's ability to detect acute disordersKim et al. (2023). Performance issues often arise from biased models trained with imbalanced data, where the abundance of non-pathological data overshadows the rare pathological instances. Despite their rarity, these instances are highly significant as they contain crucial information. Detecting such rare events requires advanced methods that account for their infrequency, unique features, and positive-negative evidences, since the true rare event distribution cannot be fully captured from these sparse observations using purely statistical methods Abubakar et al. (2024). Rare event detection is especially important in scenarios like cancer detection or identifying the specific area where disease onset occurs Che et al. (2023), and it remains a challenging task.

To address OOD generalization challenges, researchers have explored techniques such as domain adaptation (DA) and multi-source domain generalization (MSDG), where data from multiple target domains or centers are used during the training process Hu et al. (2023). These techniques have shown some success in improving DL performance on testing datasets derived from different demographics, scanners, or centers. However, DA / MSDG methods often come with high costs for acquiring target-domain data and pose privacy concerns due to data redistribution from multiple sources, making them less practical for clinical use Hu et al. (2023). In contrast, single domain

generalization (SDG) offers a more feasible approach, aiming to generalize from a single source dataset Yan et al. (2023); Vidit et al. (2023). While extensively researched in the context of image classification, to the best of our knowledge, no prior work has addressed the challenge of SDG for rare events image classification, let alone with limited data in the medical domain.

To improve SDG for rare events, leveraging expert knowledge such as clinical opinion on disease signature, that remains consistent across domains, distinguishes well between classes, and provides valuable insights for intra-class variability, has immense potential. However, in the medical field, expert knowledge tends to be fuzzy, subjective, and open to interpretation Boerwinkle et al. (2017); Hunyadi et al. (2015). Additionally, the prevalence of noisy data (measurement artifacts) in medical datasets can lead to overlaps with other classes, both visually and knowledge wise (discussed in Section 3.1 and 3.2), exacerbating the challenge Kamboj et al. (2024a); Boerwinkle et al. (2017). Consequently, many knowledge-based systems used in the medical domain have shown high rates of false positives (FPs) and false negatives (FNs) Kamboj et al. (2024b;a); Banerjee et al. (2023); Nandakumar et al. (2023).

In this paper, we introduce a fundamentally different approach to SDG for rare events in medical domain. We utilize non-data driven, discriminative expert knowledge and carefully integrate it with DL using pre-trained large vision model (LVM) and class-wise entropy. Integrating DL with domain expertise to achieve SDG for rare event detection in medical domain requires addressing several challenges simultaneously: extremely limited data, highly imbalanced datasets due to event rarity, overlapping data and knowledge rules between classes, less inter-class variability and high intra-class variability. Medical datasets, which are typically smaller, and more costly to collect and annotate compared to generic image datasets pose a unique challenge due to their limited labeled data. This scarcity complicates the use of traditional methods like data augmentation or collecting diverse samples to resolve data overlap issues. Consequently, a blind implementation of knowledge-DL techniques may weaken integration and fail to incorporate the nuanced, subjective knowledge and address rare event detection and SDG issues essential for effective medical applications. In this paper, we demonstrate how the careful fusion of data-centric DL techniques with non-data-centric knowledge can effectively handle class imbalance, data / knowledge overlap, and limited data availability, leading to effective SDG performance in rare event detection.

We consider data from two centers, $A$ and $B$, and perform the following experiments: a) across trial validation, where our approach is trained using data from center $X \in A, B$ and tested using data from center $Y \in \{A, B\} : X \neq Y$, and b) aggregate trial cross validation, where both training and test data is sampled from both centers. Our extensive experiments and comparisons with state-of-the-art DL techniques show that we achieve the average F1 score of 90.1%, while maintaining the overall F1 score of greater than 85% across datasets indicating robust SDG performance across centers. Additionally, aggregate trial outperformed SDG performance with F1 score of 94.7%, demonstrating the ability to capture domain-invariant features and generalize performance effectively.

To summarize our contributions: i) We introduce a novel algorithm, RareSaGe, that integrates domain invariant expert knowledge with DL for rare events detection and SDG, using LVM and class-wise entropy, overcoming the challenge of limited and imbalanced data, as well as overlapping data and knowledge. This integration involves center invariant knowledge of experts, and merges complementary methodologies (DL + knowledge) to address different facets of the classification tasks involving rare events and overlapping information. ii) Extensive experiments and comparisons show that DL models, when augmented with expert knowledge and designed to break information overlap, effectively detect rare events in medical imaging data from disparate sources, thereby demonstrating enhanced generalizability using single domain.

## 2 RELATED WORK

**Domain Adaptation and Domain Generalization**. DA aims to align the source domain distribution with a specific target domain, while MSDG leverages multiple source domains with varying statistical distributions to build models that generalize well to unseen target data Hu et al. (2023). Although these techniques have shown some success in image classification of balanced datasets, they require either target-domain data or multiple source data in advance, which poses significant challenges in the medical domain Hu et al. (2023). SDG involves using labeled data from a single source domain to learn higher-level domain-invariant features, such as language structures or

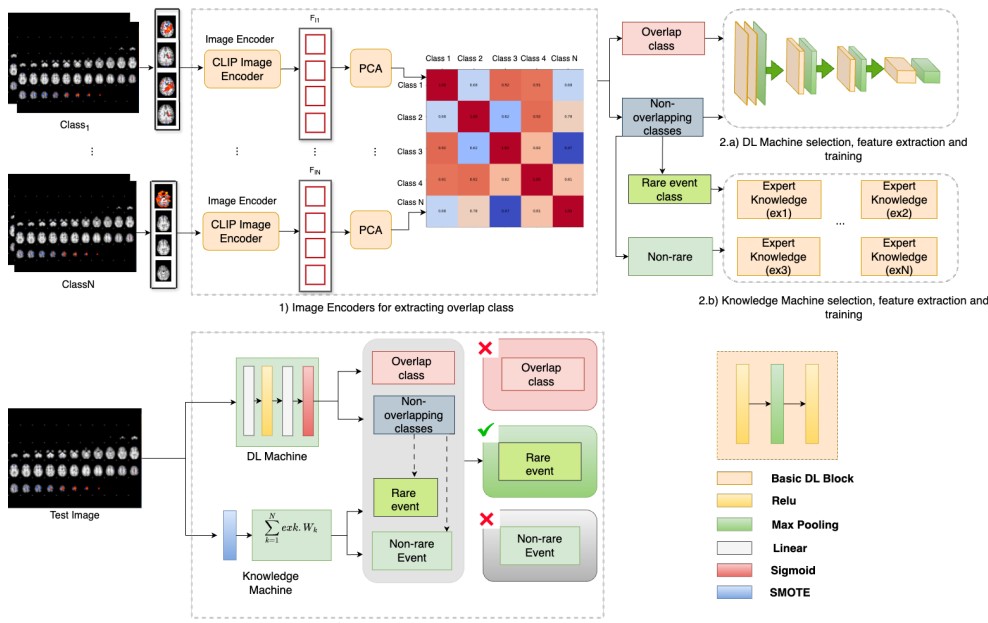

Figure 1: **Model Schematic: 1) Overlap Class Extraction:** CLIP image encoders extract features from raw class data to identify most similar class to the rare event class (Definition 3.1). **2.a) DL Machine:** This selects a DL machine to process data from the overlap and non-overlap classes, extract features, and train the model. **2.b) Knowledge Machine:** This extracts features from the rare event and non-rare event classes, and trains and select a quadratic optimization model. **3) Label Predictor:** Each test image is first classified by the DL model as overlap or non-overlap. Non-overlapping images are then classified by the knowledge model as rare or non-rare.

concept models Vidit et al. (2023). While SDG has been investigated for image classification, existing methods have notable limitations. They typically require a balanced dataset across classes and struggle in situations with limited data, especially for rare event classification in the medical domain. This scenario is particularly challenging due to the scarcity of overall data and even more limited instances of rare event, making it difficult to extract domain-invariant features that heavily depend on data availability Hu et al. (2023)Li et al. (2023)Chen et al. (2023b)Kim et al. (2023)Yang et al. (2023). Another approach for detecting rare events involves using image-level augmentation techniques in DL models. However, recent studies suggest that these methods are even less effective for both image-level and feature-level generalization Hu et al. (2023). This ineffectiveness stems from the fact that augmented images or features originate from the same distribution as the source domain and fail to capture the diverse representations of rare event data. Consequently, these methods often miss important variations due to limited data availability and thus fail at domain generalization.

**Knowledge and DL**. Using knowledge with DL for enhanced classification often involves preparing knowledge-based data for training, such as encoding expert knowledge to create features Banerjee et al. (2023), or employing domain-specific image augmentation techniques Hu et al. (2023). However, since clinical knowledge is often overlapping across classes, such as similar signal power spectrum thresholds representing two different classesBoerwinkle et al. (2017), such techniques may not have sufficient discrimination. Other techniques incorporate domain-specific knowledge directly into model training usually through loss functions to regularize the model better. For instance, in physics-guided neural networks Daw et al. (2021), where equations representing domain knowledge are added to the DL loss. In medical domain, where knowledge is not crisp and difficult to express in equations, this presents a limitation. Techniques that refine final classification decisions using domain knowledge, such as Knowledge-Enhanced Neural Networks, modify neural network predictions based on logical constraints derived from domain knowledge Daniele & Serafini (2019). However, these systems struggle when data is uncertain, or knowledge is incomplete or inconsistent, making manual design difficult and ineffective. Additionally, they enhance predic-

tions by using symbolic rules based solely on the output classes, without capturing any intra-class variability Daniele & Serafini (2019).

In this paper, we develop a model that doesn't require any target or multi-domain data for training. We leverage knowledge to address the challenges posed by limited and imbalanced data of rare class, and address knowledge and data overlap while integrating DL and knowledge to enhance generalizability using data from just a single domain.

## 3 METHOD

### 3.1 PRELIMINARIES AND PROBLEM STATEMENT

"Rare events are extremely infrequent events whose characteristics make them or their consequences highly valuable. Such events appear with extreme scarcity and are hard to predict, although they are expected eventually" Sokolova et al. (2010). In this paper, we consider *rare event* to be a phenomenon which has four properties: a) **Discrimination**: observations of the phenomenon have distinctive characteristics than other observations, b) **Scarcity**: the phenomenon has less number of observations in the process, c) **Significance**: each observation of the phenomenon has much more information content than other observations of the process, d) **Overlap**: each observation of the phenomenon has several characteristic features that exhibit significant overlap or high similarity within the feature embedding space, independent of class labels. This means that the embedding space has not been exposed to the specific classes of the domain in question and has not been trained to differentiate between them.

Observations of the process can be categorized into a finite number $(m)$ of classes common across domains. A *rare class*, $c_r \in \{c_1, \ldots, c_m\}$ is defined as a collection of *rare events* across all domains.

**Rarity Quantification:** We quantify rare class using class wise entropy metric Li et al. (2019). We define distance $dist(x_i, x_j)$, between representations $x_i$ and $x_j$ of two observations $y_i$ and $y_j$ in dataset $Y$ with $n$ observations in a given domain $z \in \{1 \ldots q\}$ ($q \geq 2$ domains) with distribution function $\mathcal{D}_z(c_k, \omega_k)$ for any class $c_k$ with class specific parameter set $\omega_k$. For each observation $y_i$ of a rare class $c_r$, we define $Q(x_i)$ as the set of all observations $y_j$ such that $x_j, x_i \in c_r$, and $y_j$ is in the $K$ nearest neighbor set of $y_i$ using the distance metric $dist(x_i, x_j)$. The set $Q(x_i)$ defines density of $y_i$, $\lambda(x_i)$ in Eqn. 1.

$$\lambda(x_i) = \frac{1}{|Q(x_i)|} \sum_{j=1}^{|Q(x_i)|} \frac{1}{\text{dist}(x_i, x_j)}, \tag{1}$$

where $|Q(x_i)|$ is the number of observations in the set $Q(x_i)$. The class average density of an observation $y_i \in c_r$ is given by Eqn. 2.

$$\gamma(x_i) = \frac{\lambda(x_i)}{\sum_{j=1}^{|c_r|} \lambda(x_j)}, \tag{2}$$

The class entropy for $c_r$ is defined using Eqn. 3,

$$\theta_r = \sum_{i=1}^{|c_r|} (-\gamma(x_i) \log_2 \gamma(x_i)). \tag{3}$$

In this paper, we define a class $c_r$ as a rare class if, for all domains $z \in \{1 \ldots q\}$, $\theta_r^z$ ($\theta_r$ for a domain $z$) falls outside $2\sigma_\theta^z$ range from the mean value $\theta_M^z$ of $\theta$ across all classes in domain $z$, where $\sigma_\theta^z$ is the standard deviation of the class entropy values across all classes, i.e.
$\theta_r^z > \theta_M^z + \sigma_\theta^z$ *or* $\theta_r^z < \theta_M^z - \sigma_\theta^z$,
where, $\theta_M^z = \frac{\sum_{i=1}^m \theta_i^z}{m}$ and $\sigma_\theta^z = \sqrt{\frac{\sum_{i=1}^m (\theta_i^z - \theta_M^z)^2}{m}}$.

[SDG Rare class problem definition] Consider at least two mutually exclusive datasets $Y_1$ and $Y_2$ ($Y_1 \bigcap Y_2 = null$) from two different domains, such that each instance from each domain belong to a unique class out of the set of $m$ classes $C = \{c_1 \ldots c_m\}$. Class $c_r$ is rare according to Definition 3.1 in both $Y_1$ and $Y_2$. Using only observations from dataset $Y_1$, determine if a test data $t_y \in Y_2$ is a member of $c_r$ or not.

**Rationale for Definition 3.1:** The fundamental reason ML techniques may fail in solving Problem Definition 3.1 is that there are not enough information available in the rare class observations to estimate the true distribution $\mathcal{D}_z(c_r, \omega_r)$, where $\omega_r$ is the parameter of the distribution for class $c_r$ Abubakar et al. (2024). The lower bound of the error in estimating the parameters of a distribution using an unbiased estimator is given by the Cramer Rao Lower Bound (CRLB) as:

$$e_{\omega_r} \geq 1/I(\omega_r), \tag{4}$$

where $I(\omega_r)$ is the Fischer information Lin et al. (2005). Barron et al. Barron (1986) provides a close form relationship between class entropy $\theta$ computed using Eqn. 3 and Fischer information $I(\omega)$, where as $\theta$ increases, $I(\omega)$ should decrease. According to the CRLB theory, this implies that as $\theta$ increases $e_\omega$ increases. Hence, for the rare class, since class entropy is higher, the error in estimating the distribution by an unbiased estimator is higher, i.e., $e_{\omega_r} > e_{\omega_i}, \forall i \in \{1 \dots m\}, i \neq r$.

The above-mentioned result implies that a direct ML/DL based approach towards solving SDG for rare class problem in Definition 3.1 may not be successful, since it will have higher error in detecting rare class than other classes.

## 3.2 GENERIC SOLUTION FOR SDG RARE CLASS PROBLEM

**Solution overview:** In accordance with rare class definition, we first isolate the rare class from the dataset. Once identified, we extract features from the raw data of both the rare and other classes using CLIP. The class whose features most closely resemble those of the rare class is then identified as the "overlap class", representing visual similarities that complicate automated distinction between these classes. To enhance the detection of the rare class, we address overlap class discrimination using DL models, while the non-overlap classes and rare class is distinguished using knowledge-based models to compensate for data scarcity. Finally, to determine the most effective DL model for overlap class classification, and the best knowledge-based model for rare and non-rare class distinction, we use a class-wise entropy-based technique.

**Solution details:** Our approach to solving the SDG rare class problem in definition 3.1 involves integrating expert knowledge with DL techniques. First, we consider the source domain $Y_1$ and isolate the rare class from the other available classes in the raw data, following the criteria outlined in Definition 3.1. For this purpose, we need a representation $x_i$ of the observations $y_i \in Y_1$. For DL, we employ large vision model (LVM) CLIP to extract features from the raw data, independent of class information (Fig. 1) Radford et al. (2021). We then apply Principal Component Analysis (PCA) to the CLIP extracted features to obtain the significant representations. For expert knowledge, we utilize symbolic AI based representations highlighted in Section 3.4.2. Utilizing PCA representation, we compute the similarity between the class agnostic feature embeddings of the rare class and each of the classes in $C$, referring the most similar class to the rare class as the overlap class. The overlap $c_o = argmin(\theta_i)$ is then chosen to be classified using DL techniques, based on a machine selection strategy, in a two class classification problem, $c_o$ v.s. $\neg c_o$. For rare class which is embedded in the $\neg c_o$ class, we then utilize machine selection strategy to derive the best knowledge based quadratic optimization (QO) machine to identify the rare class $c_r$ from the $\neg c_r$ class.

**Machine orchestration strategy:** We assume that there is a set of trained DL/ML classifiers $\mathcal{M}_{\mathcal{DL}}$ for the overlap and non-overlap classes' learning, and a set of trained QO based classifiers $\mathcal{M}_{\mathcal{QO}}$ for the knowledge-based rare class and non-rare classes (normal events) learning. Each classifier $M_d \in \mathcal{M}_{\mathcal{DL}}$ or $M_d \in \mathcal{M}_{\mathcal{QO}}$, classifies instances of $Y$ with a label from the set $S^{M_d} \subset 2^C$, such that each label $s_k^{M_d} \in S^{M_d}$ meets the following rules:

$$\forall k, l \in \{1 \dots |S^{M_d}|\}, k \neq l, s_k^{M_d} \bigcap s_l^{M_d} = \phi \text{ mutual exclusion} \tag{5}$$

$$\bigcup_{k=1}^{|S^{M_d}|} s_k^{M_d} = C \text{ completeness}$$

$$\forall s_k^{M_d} \in S^{M_d} \exists Z \subset C : s_k^{M_d} = \bigcup_{z \in Z} z \text{ union of original labels}$$

$\phi$ is the null set. A classifier $M_d$ represents the raw data $y_i \in Y$ in some latent representation space $x_i \in X$ using a discriminative feature function $\mathcal{F}_{M_d}$. The same function can be used to represent each raw data in the original class set $C$. This representation is used in Eqn. 1

to compute a new $\lambda^{M_d}(x_i)$ by using $x_i = \mathcal{F}_{M_d}(y_i)$. Following the entropy calculation in Eqn. 3 we can derive the entropy $\theta_r^{M_d}$ for each classifier $M_d$ and select the classifier with the least entropy.

**RARE event detection with Single domAin GEneralization (RareSaGe) overview:** Algorithm 1 is used to derive a classifier for the rare class $c_r$. It takes three configuration parameters: a) mean class entropy $\theta_M$ for domain $Y_1$, b) standard deviation of $\theta$ in domain $Y_1$, $\sigma_\theta$, and c) confidence threshold $t_c$, used to set preferences for classifiers typically based on computational resources and memory requirements for that classifier. Algorithm 1 also takes the training data and a set of classifiers $\mathcal{M}$ as input. It then runs the following steps:

**Step 1:** Extract rare class according to Definition 3.1

**Step 2:** Identify the overlap class $c_o$ most similar to $c_r$ in the class label agnostic embedding space.

**Step 3:** Determine the best possible machine among DL and expert knowledge to classify $c_o$ and $\neg c_o$. This is done by evaluating the Entropy for each machine on the data of domain $Y_1$ with classes $c_o$ and $\neg c_o$. The best classifier is the one with the lowest entropy.

**Step 4:** From $\neg c_o$ instances, divide them into $c_r$ and $\neg c_r$ classes.

**Step 5:** Repeat the step 3, this time for $c_r$ and $\neg c_r$ class to find a knowledge based classifier $\mathcal{M}_{\mathcal{QO}}$.

**Step 6:** Combine the $\mathcal{M}_{\mathcal{DL}}$ and $\mathcal{M}_{\mathcal{QO}}$ using the confidence threshold $t_c$.

---

**Algorithm 1** RareSaGe Algorithm

---
**Input:** Raw data $Y$ with $n$ classes $\{C_1, C_2, \ldots, C_n\}$, $\theta_M$, $\sigma_\theta$, confidence threshold $t_c$, set of classifiers $\mathcal{M}_{\mathcal{DL}}$ and $\mathcal{M}_{\mathcal{QO}}$.

1: Sample set $\Psi = Y$
2: **while** $\Psi$ is not empty and significant change in validation accuracy **do**
3:     Compute proportion $\theta_i$ of each class $C_i$, where $i \in \{1, 2, \ldots, n\}$
4:     Identify rare class $C_{rare}$ such that $\theta_{rare}$ satisfies Definition 3.1.
5:     Compute the overlap class $c_o$ utilizing the highest similarity with $C_{rare}$ in a class agnostic embedding space representation.
6:     Divide the dataset into $c_o$ and $\neg c_o$ classes
7:     **for** each classifier $M_d \in \mathcal{M}_{\mathcal{DL}}$ **do**
8:         Compute $\theta(M_d)$ on the set $\Psi$ for $c_o$ and $\neg c_o$
9:     **end for**
10:    Choose classifier with minimum $\theta$: $M_d \leftarrow \arg \min_{M_d} \theta(M_d)$.
11:    Divide $\neg c_o$ into $c_r$ and $\neg c_r$
12:    **for** each classifier $M_d \in \mathcal{M}_{\mathcal{QO}}$ **do**
13:        Compute $\theta(M_d)$ on the set $\Psi$ for $c_r$ and $\neg c_r$
14:    **end for**
15:    Choose classifier with minimum $\theta$: $M_d \leftarrow \arg \min_{M_d} \theta(M_d)$.
16:    Compute confidence scores for classifiers $M_{DL}$ and $M_{QO}$
17:    Choose classifier $M_{QO}$ with score $> t_c$ for rare event labelling
18:    **Return** Final labels: Overlap Class, Rare Class, Non-rare Class
19: **end while**=0

---

We show synergistic benefits of integrating expert knowledge encoding and DL in generalized rare event identification from medical imaging through a case study on fMRI based pre-surgical screening of patients with drug resistant epilepsy. It is essential to locate Seizure onset zone (SOZ) from resting state fMRI (rs-fMRI) which can be then surgically altered to reduce seizure frequency Nandakumar et al. (2023) Kamboj et al. (2024a) Banerjee et al. (2023) Hunyadi et al. (2015).

### 3.3 RARE SOZ IDENTIFICATION PROBLEM STATEMENT

In the standard pre-surgical screening process, rs-fMRI is collected from patients, resulting in 4D spatio-temporal data. Independent component analysis of rs-fMRI results in decomposition of the signal into: noise spatial and temporal independent components (IC), capturing activation artifacts due to head movement or measurement noise, resting state network (RSN) spatial and temporal ICs capturing normal brain activity, and SOZ spatial and temporal ICs capturing onset of seizure activity (rare class). fMRI typically yields between 100 to 200 spatial independent components (ICs), each with both spatial and temporal signals, per patient. Only a small fraction (5-10%) of them belong to SOZ ICs Kamboj et al. (2024a) Banerjee et al. (2023).

**Given:** A dataset $Y_A$ from center $A$. $Y_A$ consists of a set of $N$ spatial ICs, and $N$ temporal ICs from fMRI consisting of three classes: 1. Noise class ($\sim$55%, ICs primarily affected by measurement

noise), 2. Resting state network class ($\sim$40%, with activation primarily affected by normal brain function), and 3) SOZ (rare event) class ($< 5\%$, with activation affected by seizure onset). Given another dataset $Y_B$ consisting of a set of $M$ spatial and temporal ICs.

**Input:** A spatial and temporal IC $y_b \in Y_B$.

**Find:** Whether the $y_b$ is class SOZ or not. .

### 3.4 OUR SOLUTION

Fig. 1 shows the overall architecture of expert knowledge integration with DL for rare event detection.

#### 3.4.1 DEEP LEARNING FOR NOISE

DL techniques, including Vision transformers (ViT), language vision models (LVM)s Radford et al. (2021), 2D-CNNs, and transfer learning using pre-trained models are effective at capturing intricate spatial patterns and features within images He et al. (2016). Given the prevalence of noise class instances in medical imaging datasets, often constituting over 50% of the data Boerwinkle et al. (2017); Banerjee et al. (2023); Kamboj et al. (2024a), DL can leverage its capability to detect subtle spatial patterns and features that distinguish noise from other classes.

#### 3.4.2 EXPERT KNOWLEDGE ON RS-FMRI

We utilize two types of expert knowledge on rs-fMRI data:

**a) Anatomical knowledge:** This pertains to the spatial locations of anatomical brain parts crucial for SOZ recognition. These locations can be extracted employing established image processing algorithms. The utilized specific locations are:

**1. Brain periphery:** This is the brain boundary obtained through Sobel filter based contour detection mechanism Banerjee et al. (2023).
**2. Gray matter:** Extraction is achieved through a combination of three methods: Global Probability of Boundary, Oriented Watershed Transform, and Ultrametric Contour Map as detailed in Banerjee et al. (2023). This method is applied on the gray scale image to extract the gray colored areas of the brain.
**3. White matter:** Similar extraction method as gray matter, focusing on white regions within the grayscale rs-fMRI image.
**4. Vascular regions:** Areas of blood vessels in the brain, extracted through slice processing techniques detailed in Banerjee et al. (2023).
**5. rs-fMRI activation:** Activation in the rs-fMRI image is obtained through DBSCAN, with min-Points parameter set to 2 voxels (where a voxel represents the smallest activation unit, typically a 3 pixel $\times$ 3 pixel image segment), and a neighborhood distance set to 1 voxel. All activation clusters below 135 voxels are disregarded as weak.

**b) Expert knowledge on specific rare event:** This encompasses knowledge about rs-fMRI activation patterns observed for SOZ, compiled from the works of Hunyadi et al. Hunyadi et al. (2015) and Boerwinkle et al. Boerwinkle et al. (2017). SOZ specific knowledge is expressed as logical connectives of atomic propositions on the relative location of activation with respect to the anatomical regions of the brain as detailed in the following list:

$p_1$ : Presence of a single activation cluster inside brain boundary, computed by counting the clusters of size greater than 135 voxels, completely inside the brain peripheral contour.
$p_g$ : Activation primarily located in gray matter, calculated by the percentage of voxels within the activation lying inside the gray matter contour using the winding number algorithm Jacobson et al. (2013).
$p_s$ : Sparse representation of the blood oxygen level dependent (BOLD) signal in the sine domain, computed using Gini Index on the BOLD signal frequency response of each IC Hunyadi et al. (2015).
$p_a$ : Activelet domain representation of the BOLD signal is sparse. This is also computed by first performing activelet transform and then computing sparsity using Gini Index Hunyadi et al. (2015).
$p_w$ : Activation overlaps with white matter. This is computed using the same method as $p_g$, but on white matter contours.

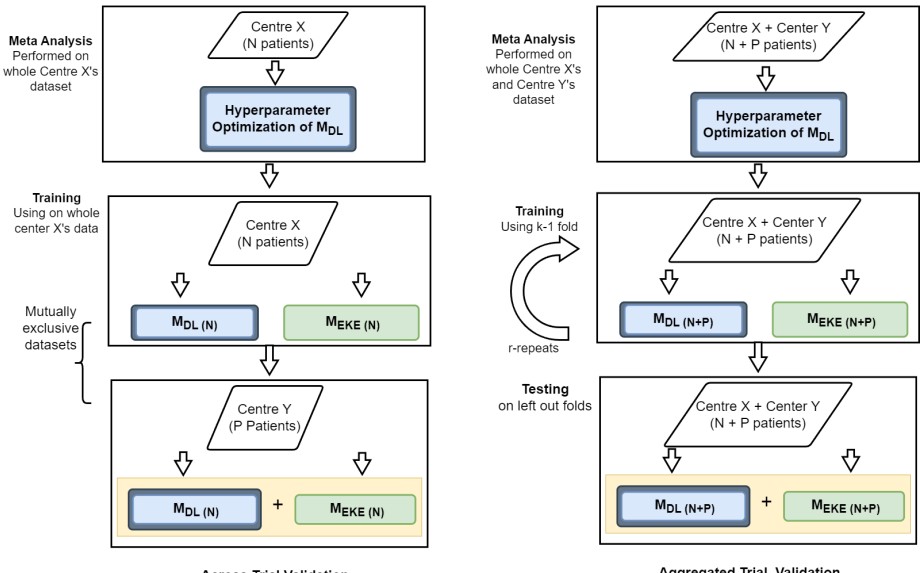

Figure 2: Experiments performed to evaluate generalizability of rare event detection: A) Across trial validation with training in one center $X$ and testing on the other $Y \in \{A, B\} : Y \neq X$, and B) Aggregate trial with 5 fold 3 repeats validation.

$p_v$ : Activation overlaps with vascular regions. This is computed using the same method as $p_g$, but on vascular contours.

Utilizing the aforementioned atomic propositions, the expert knowledge on SOZ rare events can be encapsulated in the following first-order logical formula:

$$\kappa_{SOZ} = p_1 \wedge \neg p_s \wedge p_a \wedge [p_g \wedge (\neg p_w \vee (p_w \wedge p_v))], \tag{6}$$

which represents the knowledge that SOZ activations primarily have one big activation cluster, the BOLD signal for SOZ ICs are not sparse in sine domain but are sparse in activelet domain, activation should primarily be in gray matter with no white matter overlap, or they may span across gray matter, white matter, and vascular regions.

The values of the atomic propositions in Equation 6 from the ICs are used as features in a QO machine to classify an IC as RSN or SOZ. Synthetic Minority Oversampling Technique (SMOTE) is used to generate synthetic features for SOZ, to create a balanced dataset before QO machine training Chawla et al. (2002). In this work, we choose Support vector machine (SVM) as a QO machine. This SVM with expert knowledge proposition values as features is called the Expert Knowledge Extractor (EKE). SVM was chosen as it is less susceptible to overfitting and promotes better generalization through the margin parameter and the use of a regularization mechanism, making it well-suited for small to medium-sized datasets. Additionally, SVMs with expert knowledge features offer human-interpretable models, facilitating understanding of the factors influencing classification decisions. This interpretability is valuable in the medical domain where decision-making transparency is crucial.

### 3.4.3 INTEGRATION OF EXPERT KNOWLEDGE WITH DL USING ALGORITHM 1

Following Algorithm 1 we first identify rare class. From domain $Y_A$, the cross entropy using CLIP features for Noise was $\theta_{Noise}^{CLIP} = 0.004$, for RSN was $\theta_{RSN}^{CLIP} = 0.0046$, and for SOZ it is $\theta_{SOZ}^{CLIP} = 0.026$. We identify that SOZ is the rare class since $\theta_{SOZ}^{CLIP}$ satisfies Definition 3.1. Further, the cosine similarity of CLIP features between Noise and SOZ was $0.78$ while that between RSN and SOZ was $0.74$. Hence, the Noise class is determined to be the overlap class. We divide the dataset into $NOISE$ and $\neg NOISE$ class. Since we do not have knowledge based machines for Noise,

Table 1: Performance results of across-trial validation - single domain generalizability for rare event detection. Values in parenthesis represent standard deviation.

| Method | Accuracy | Precision | Sensitivity | F1-score | Average F1-score |
|---|---|---|---|---|---|
| Pre-trained CNN Train A, Test B | 67.7% | 87.5% | 75.0% | 80.7% | 49.1% |
| Pre-trained CNN Train B, Test A | 9.6% | 62.5% | 10.2% | 17.5% | |
| Pre-trained ViT small Train A, Test B | 64.5% | 86.9% | 71.4% | 78.4% | 77.2% |
| Pre-trained ViT small Train B, Test A | 61.5% | 91.4% | 65.3% | 76.1% | |
| Pre-trained ViT base Train A, Test B | 45.1% | 82.3% | 50.0% | 62.2% | 60.8% |
| Pre-trained ViT base Train B, Test A | 42.3% | 88.0% | 44.8% | 59.4% | |
| ViT trained from scratch, Train A, Test B | 12.9% | 57.1% | 14.2% | 22.7% | 46.3% |
| ViT trained from scratch, Train B, Test A | 53.8% | 90.3% | 57.1% | 70.0% | |
| LVM fine tuned-contrastive loss, Train A, Test B | 64.5% | 86.9% | 71.4% | 78.4% | 46.3% |
| LVM fine tuned-contrastive loss, Train B, Test A | 7.6% | 57.1% | 8.1% | 14.2% | |
| LVM fine tuned-cross entropy loss, Train A, Test B | 51.6% | 84.2% | 57.1% | 67.9% | 37.6% |
| LVM fine tuned-cross entropy loss, Train B, Test A | 3.8% | 40.0% | 4.0% | 7.3% | |
| Knowledge based system, Train A, Test B | 83.8% | 89.6% | 92.8% | 91.2% | 78.9% |
| Knowledge based system, Train B, Test A | 50.0% | 89.6% | 53.0% | 66.6% | |
| **DL+EKE Train A, test B** | **90.3%** | **90.3%** | **100%** | **94.9%** | **90.2%** |
| **DL+EKE Train B, test A** | **75.0%** | **92.8%** | **79.5%** | **85.6%** | |
| **DL+EKE Train A Children, test B Adults** | **90.9%** | **90.9%** | **100%** | **95.2%** | **95.2%** |

we utilize DL techniques to classify $NOISE$ and $\neg NOISE$. The $\neg NOISE$ class is then used to determine SOZ. Here we have SOZ specific discriminative expert knowledge and we use the knowledge based machines to identify SOZ. We integrate expert knowledge with DL in the post processing step (Fig 1, Label Predictor). The DL first classifies IC as Noise (overlap) or non-Noise (non-overlap) class. Simultaneously, the EKE classifies IC as SOZ (rare) and RSN (non-rare/normal). If DL classifies an IC as noise and EKE categorizes it as SOZ with a confidence score surpassing a selected threshold of 0.9, the IC is labeled as SOZ; otherwise, it retains noise label. For ICs labeled as non-noise by DL, the EKE label of RSN or SOZ is selected as the final label.

## 4 EXPERIMENTS AND RESULTS

### 4.1 DATASETS

Two datasets were collected from independent centers, A and B, in compliance with IRB protocols and cross-university agreements. Center A includes 52 pediatric patients (23 Male, 29 Female, ages 3 months to 18 years) with 5,616 images (2,873 Noise, 2,427 RSN, 316 SOZ), acquired using a 3T Philips Ingenuity scanner. Center B includes 31 patients (14 Male, 17 Female, ages 2 months to 62 years) with 2,364 images (1,090 Noise, 1,072 RSN, 202 SOZ), acquired using a Siemens MAGNE-TOM Prisma FIT scanner. Code and data is available at https://anonymous.4open.science/r/SOZ-Localization-using-DL-andexpert-knowledge-D7C3/. Detailed data acquisition methods are provided in the supplementary document.

### 4.2 IMPLEMENTATION DETAILS

To evaluate generalizability, we perform two experiments motivated from Chekroud et al. (2024):
**A) Across trial validation:** Here, $M_{DL(N)}$ and $M_{EKE(N)}$ is trained on data of center $X$ with $N$ patients and tested on dataset of the other center $Y \neq X$ with $P$ patients (Fig. 2). This trial evaluates the generalization performance for the worst case distribution shift.

**B) Aggregated trial validation:** Data from all centers combined ($A \bigcup B$) was evaluated using 5-fold, 3-repeats validation. Each center's data was split into 5 subsets, with models trained on 4 subsets and tested on the remaining one (Fig. 2). This process was repeated 3 times with randomized fold sampling to assess the technique's ability to learn domain-invariant features from multi-center data. Experiments were conducted on an Intel Xeon CPU (8 cores, 93 GB RAM) and an NVIDIA Quadro RTX 5000 GPU (16 GB memory).

### 4.3 EVALUATION METRICS, RESULTS AND INSIGHT

**Evaluation metrics:** Trials are evaluated with standard metrics used in Hunyadi et al. (2015); Kamboj et al. (2024b). For SDG we: a) Assess the state-of-the-art DL techniques for SOZ (rare) identi-

fication in across trial experiments, b) compute the difference between DL+EKE accuracy on SOZ identification and baseline comparators across trial experiments, and c) finally show the aggregate-trial performance of our technique for SOZ identification.

Table 2: Aggregate-Trial Validation Results for DL+EKE.

| Repeat | Accuracy | Precision | Sensitivity | F1 score | Mean F1 score |
|---|---|---|---|---|---|
| Repeat 1 | 91.6% (6.9) | 92.8% (7.8) | 98.7% (2.8) | 95.4% (3.8) | 94.7% (3.7) |
| Repeat 2 | 89.1% (6.8) | 92.5% (5.0) | 95.8% (3.7) | 94.1% (3.7) | |
| Repeat 3 | 90.3% (7.0) | 92.6% (5.0) | 97.2% (3.8) | 94.8% (3.8) | |

**Results:** Table 1 presents the results of state-of-the-art DL techniques and our methodology on the across-trial validation test, as discussed in Section 4.2. The DL techniques used for comparison included: a pre-trained CNN using "VGG-16," pre-trained ViTs using "vit-small-patch16-224" and Google's "vit-base-patch16-224," ViTs trained from scratch with hyperparameters optimized using Optuna, the LVM CLIP Radford et al. (2021), and knowledge-based systems. All pre-trained models were fine-tuned on data from specific centers, with loss updated using class weights to address dataset imbalances due to rare events, and early stopping strategies applied to prevent overfitting. For LVM, we conducted two evaluations: one fine-tuning with contrastive loss and another with cross-entropy loss, both using the "ViT-B/32" model. Our methodology utilized a 2D-CNN as a DL machine, which performed best for evaluating overlap (noise) and non-overlap (non-noise) classes (Fig. 1). In the across-trial experiment, our DL+EKE method achieved an F1 score of 90.2%, consistently maintaining F1 scores above 85.0% for both centers. This significantly outperforms other baseline comparators, highlighting our technique's generalizability to unseen data and its ability to learn domain-invariant expert knowledge without overfitting. Furthermore, we conducted another distribution shift analysis by evaluating across-trial cross-validation based on age. Here, we trained the DL+ EKE model on all of Center A's data, spanning children (1 month - 18 years), and tested it solely on adults from Center B (18 years to 62 years). The F1 score of 95.2% attests to the model's generalizability not only to unseen data during training but also to data with entirely distinct distributions. Additionally, in the aggregated trial validation, where models were exposed to more data from both centers, Table 2 demonstrates that our technique achieves the best results, with an average F1 score improvement of 4.5%.

**Insights:** Our across-trial experiments reveal that state-of-the-art DL techniques struggled to differentiate between NOISE and SOZ classes, as indicated by CLIP similarity results, leading to suboptimal performance in rare event detection. A knowledge-based system trained on Center A accurately identified SOZ characteristics in Center B with high precision. However, when trained on Center B and tested on Center A, precision remained stable, but sensitivity dropped significantly, highlighting an increase in FNs due to patient variability. Integrating DL for overlap class separation with knowledge-based methods improved both FPs and FNs, resulting in a more generalized performance. The aggregate trial achieved a superior solution with a 94.7% F1 score for SOZ detection.

## 5 CONCLUSION

Integrating expert knowledge with data-driven DL enhances SDG performance for rare event detection across domains. The results suggest that expert knowledge is inherently domain-invariant, as experts are trained across various care centers and encounter diverse patient demographics. By carefully combining DL with expert knowledge while addressing overlaps, we improved generalization performance. The use of pre-trained LVM, expert knowledge via atomic propositions, and DL enabled accurate classification of rare events in unseen domains. Additionally, the use of propositional logic is inherently interpretable for experts, as each proposition is tied to their contributions. From the QO machine's weight configuration, we can assess the impact of each expert's knowledge on SOZ identification, providing explanations for the model's output, a critical feature for trust development in medical applications, and a key area for future research.

## 6 ETHICS STATEMENT

We ensured adherence to ethical standards by utilizing de-identified medical imaging data with full approval from the Institutional Review Board (IRB). Our methodologies were designed to minimize potential biases, uphold fairness, and safeguard privacy. No conflicts of interest or harmful applications arise from this work, and all legal and ethical guidelines were followed.

## 7 REPRODUCIBILIY STATEMENT

To ensure reproducibility of our results, we have made all relevant materials available. The anonymous link to our source code, which includes the implementation of the proposed model and training details is provided. Additionally, all hyperparameters, evaluation metrics, and experimental settings are described in the supplementary document.

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

## A  APPENDIX

You may include other additional sections here.

