1. **The deep learning techniques we implemented for *NOISE* and $\overline{NOISE}$ ICs classification (Table I):**

**Convolutional Neural network (CNN):** A 2D CNN was used with tuned hyperparameters. Details are mentioned in the manuscript as this deep learning technique was used for *NOISE* and $\overline{NOISE}$ IC classification.

**Transfer Learning:** We used the VGG16 pre-trained model as a feature extractor and defined our own classifier model. We used the pre-trained weights of 'imagenet' in Keras, we added flatten, dense and output layers after the last layer in VGG16. The dense layer's number of neurons were tested for 5 different values: 192, 256, 512, 712 and 1024, out of which 512 gave the best accuracy. Fine-tuning the VGG model by freezing a few of its layers did not give good results.

**Vision Transformer:** We implemented Vision Transformer (ViT) to evaluate its performance for *NOISE* and $\overline{NOISE}$ IC classification. The hyperparameters were obtained using "Optuna", and to avoid gradient explosion and gradient vanishing issues, we implemented gradient clipping and batch normalization respectively.

**Table I: Deep learning results for *NOISE* and $\overline{NOISE}$ ICs classifications.**

| Technique | Accuracy | Precision | Sensitivity | Specificity |
|---|---|---|---|---|
| CNN | 80.3% | 81.8% | 76.2% | 83.4% |
| Transfer Learning | 75.5% | 76.2% | 77.8% | 73.1% |
| ViT | 69.01% | 69% | 71% | 67% |

As CNN reports the best classification results, we used this deep learning technique for noise elimination.

2. **Data description:**

Resting state fMRI data was acquired from 2 independent centers: Centre 1 and Centre 2 using IRB process and cross-university agreement. Center 1 dataset consists of 52 patients, age 3 months – 18 years (all children), 23 Male and 29 Female. The MRI images were acquired using a 3T MRI, Ingenuity Philips Medical system with a 32-channel head coil. The rs-fMRI parameters were set at TR 2000ms, TE 30 ms, matrix size 80 x 80, flip angle 80°, number of slices 46, slice thickness 3.4 mm with no gap, in-plane resolution 3x3 mm, interleaved acquisition, and number of total volumes 600, in two 10-min runs, with total time of 20 mins. Centre 2 dataset consists of 31 patients with ages spanning from 2 months to 62 years (20 childnre, 11 adults), 14 Male and 17 Female. The MRI images were acquired using Siemens's MAGNETOM Prisma FIT scanner, with the following parameters configured: TR 2010ms, TE 32ms, flip angle 82°, slice thickness 4mm and spacing between slices 4.

3. **Hyperparameter settings**: We used Keras Hyperband tuner algorithm with objective of minimizing validation loss to obtain the hyperparameters of DL part. The 2D CNN architecture was used as it gave the best results for noise IC classification (Table I). CNN's hyperparameters were fine-tuned using 80% training data, reserving the remaining 20% for validation.

### i) Across_trial cross validation (Center A): The final hyperparameters were:
a) Number of convolutional layers: 3
b) Number of 3 X 3 filters in convolutional layer 1, 2 and 3: 64, 64 and 256 respectively.
c) Number of neurons in dense fully connected layer: 704.
d) Learning rate: 0.0001.
e) Dropout rate: 0.33.
f) Batch_size = 32.

### ii) Across_trial cross validation (Center B): The final hyperparameters were:
a) Number of convolutional layers: 6
b) Number of 3 X 3 filters in convolutional layer 1, 2, 3, 4, 5 and 6: 128, 16,64, 512, 512, and 256 respectively.
c) Number of neurons in dense fully connected layer: 3008.
d) Learning rate: 0.0001.
e) Dropout rate: 0.33.
f) Batch_size = 32.

### iii) Aggregated Trial validation:
a) Number of convolutional layers: 2
b) Number of 3 X 3 filters in convolutional layer 1 and 2: 16 and 64 respectively.
c) Number of neurons in dense fully connected layer: 1472.
d) Learning rate: 0.0001.
e) Dropout rate: 0.33.
f) Batch_size = 16.

**4. Closed form relation between class entropy and Fisher information:**

According to Gourieroux et al. (1995), Fisher information is proportional to the Kullback-Leibler (KL) divergence of class entropy. Considerin $\omega_r$ as a parameter of the latent space representation of $x_i$, we obtain the following equation connecting class entropy and Fisher information:

$$\frac{\partial^2 \theta_r}{\partial \omega_r^2} = I\left(\omega_r\right)$$

where $\frac{\partial^2 \theta_r}{\partial \omega_r^2}$ indicates the second derivative or divergence of class entropy with respect to the latent space representation element $\omega_r$ .

Using Equations (1), (2), and (3) given in main manuscript, and applying algebraic manipulations, we derive the following relation:

$$I(\omega_r) = \sum_{i=1}^{|c_r|} \frac{-\partial^2 \gamma(x_i)}{\partial \omega_r^2}\left(1 + log_2\big(\gamma(x_i)\big)\right) - \frac{1}{\gamma(x_i)}\left(\frac{\partial \gamma(xi)}{\partial \omega r}\right)^2$$

Further exploration of the term $\left(\frac{\partial \gamma(xi)}{\partial \omega r}\right)^2$ yields the following relation:

$$\frac{\partial^2 \gamma(x_i)}{\partial \omega_r^2} = \frac{\partial\left[\frac{1}{|Q(x_i)|}\sum_{j=1}^{|Q(x_i)|}\frac{-1}{dist(x_i,x_j)^2}\frac{\partial\, dist(x_i,x_j)}{\partial \omega_r}\right]}{\partial \omega_r}$$

This can be expanded as:

$$= \frac{1}{|Q(x_i)|}\sum_{j=1}^{|Q(x_i)|}\frac{-1}{dist(x_i,x_j)^3}\left(\frac{dist(x_i,x_j)}{\partial \omega_r}\right)^2 - \frac{1}{dist(x_i,x_j)^2}\frac{\partial^2\, dist(x_i,x_j)}{\partial \omega_r^2}$$

If we substitute $\frac{\partial\,\gamma(x_i)}{\partial \omega_r}$ from the equation above into the expression for I($\omega_r$) after simplifications for clarity, we find that:

$$I(\omega_r) \propto \frac{1}{dist(x_i,x_j)^4}$$

Since in rare classes, the distance *dist($x_i, x_j$)* is much larger and class density is very low, the Fisher information is correspondingly low, which increases the lower error limit of any unbiased estimator.

5. Anonymous code and data link: The code link has been provided in the paper. Sample data of one patient is provided from Center A.