# OpenReview forum: "Single Domain Generalization for Rare Event Detection in Medical Imaging"
_ICLR.cc/2025/Conference — Submitted to ICLR 2025_

### Official Review · Reviewer_eoxC · 2024-10-28

**Soundness:** 2
**Presentation:** 3
**Contribution:** 2
**Rating:** 3
**Confidence:** 4

**Summary:**

The authors introduce an algorithm for generalizing to unknown domains using only training data from a single known domain, "RareSaGE", which proposes to integrate domain-invariant expert knowledge with neural networks for the problem of rare class classification. Their method achieves improved performance for this task compared to several neural network- and expert knowledge-based techniques.

**Strengths:**

1. The paper focuses on a lesser known, but still seemingly important problem which deserves attention. They test on a realistic dataset comprised of medical data sampled from real centers, demonstrating real life applicability.
2. The method is seemingly fairly technically novel. The rarity quantification doesn't seem novel (they cite Li et al), but overall, the combination of expert knowledge (an old field) with modern neural network models is interesting and unusual.
3. Improvements over the compared baselines are significant for all metrics (except maybe precision) (Table 1)
4. The intuition behind the method is reasonable, and the formalism discussed does help with understanding this. The reasoning behind "rationale for definition 3.1" makes sense, i.e., explaining the challenges in working with rare classes with respect to class entropy.
5. I think the four characterizations of a rare event make sense (not sure if these are novel or not?), although discrimination and significance may be a bit redundant.

**Weaknesses:**

**Major:**
1. **Generally limited experiments (don't even begin until page 9):** I would suggest moving many of the less-important dataset, algorithm and method details into the supplementary, and filling the missing space with more experiments, especially those considering ablation studies, failure cases for your method, computational analyses, and others. The biggest issues are as follows.
    1. For a method defined in such generality, the experiments are quite limited: just a case study based on one fMRI datasets. This is a big negative, because it is unclear if the method could generalize to other problems, or if it was bespoke to (or developed specifically for) this case study. This hurts the paper's suitability for ICLR, as the larger impact on machine learning is not clear to me, which would require testing on other datasets/problem settings/etc.
    2. Moreover, while the improvements over the compared baselines are significant for all metrics (except maybe precision) the comparison/baseline models may not be sufficient:
        - I'm not an expert in domain generalization, but are the comparison models in Table 1 really proper baselines (maybe another reviewer can chime in)? Simple training or finetuning on one domain and testing on another domain (as well as the knowledge-based systems) doesn't seem like a strong, appropriate baseline. Are there better unknown domain generalization techniques which should have been tested?
        - Alternatively, what about comparing to one-class-classification methods, i.e. OOD/anomaly-detection methods? This is also a large body of work which seems suitable. At the very least, why weren't these discussed in the related work?
        - why wasn't AUC presented if sensitivity and precision were also presented? This would be a better general performance metric than accuracy for accounting for class imbalance.

2. **Lack of ablation/hyperparameter sensitivity studies:**
    1. this method has quite a few moving parts, which each could be brittle. The more of these moving parts that there are, the more ways that the method could fail when extended to new datasets/problems. It is important to consider how changing or removing one component would affect the performance, in order to judge how reliant to algorithm is on that component. However, the paper critically lacks any ablation/sensitivity studies for such hyperparameters/settings. I will discuss potential ablation studies which could be done, but I challenge the authors to think of more. Some examples:
        1. On pg. 4, a rare class is defined by a 2-sigma distance from the mean class entropy. Why/how was 2 sigma chosen? This is seemingly an important parameter, yet you do no ablation/sensitivity studies on the effect of different values for this (number of sigmas used to define a rare class)
        2. Algorithm 1 is quite complex, additionally with certain steps not quantitatively/explicitly defined.
        3. The use of PCA on clip features may have limitations, as its linearity may be too much of a restriction. Why wasn't some form of nonlinear component/degrees-of-freedom analysis tested as well?
        3. The form of the expert knowledge in the tested scenario (Eq. 6) is quite specific. Are there viable alternatives to this formulation that could have been tested on this dataset? Also, in general, what is the feasibility for converting expert knowledge into this format?

**Minor:**
1. More clear, technical method details are needed in the abstract and introduction to be clear what these contributions are. For example:
    1. In the abstract: "This paper addresses … even with limited data availability." This really lacks in explicit details on how the method actually works. how is this ranking done, specifically? is it via a novel algorithm? also, what does "focused handling" mean? and how is domain-invariant knowledge "integrated"? These questions could be gleaned from the main text of the paper, but a better level of detail could still be provided in the abstract with less vague wording.
    2. Similarly in the introduction, more technical details are needed. Your repeatedly describe the "integration" of domain invariant expert knowledge with deep learning, but I'm unclear from reading this what this actually explicitly describes.
2. Misleading claims/vague wording:
    1. In the introduction, you say "This concept is particularly crucial in the field of artificial intelligence (AI) for medicine, where the aim is to accurately diagnose new patient cases across centers". This is misleading. AI for medicine extends far beyond diagnosis, including tasks such as segmentation, registration, harmonization, etc… and that's just for images, not including AI for medicine beyond images.
    2. "In medical imaging, there are various factors …  acute disorders" also in the introduction; what are the "classes" being described here? I'm not sure if this is really intra-class variability (at least since you don't explain what the classes are here), its really just various factors that can contribute to domain shift problems in medical datasets.
    3. "In the medical diagnosis and disease detection domain, where data is often limited … variability in rare event characteristics. " in the abstract is phrased confusingly. in the first sentence, is performance poor on the data because of limited data, rare cases of interest, or both? Maybe say that the second sentence describes a problem which exacerbates the problem described in the first sentence.
3. Other missing technical details:
    1. how is feature overlap/resemblance (described in the first paragraph of Sec. 3.2 actually computed? cosine similarity of the feature vectors?
    2. Algorithm 1 shouldn't really be labeled as an "algorithm" in my opinion, because many of the steps are not defined explicitly, and left up to interpretation. More details are needed.
    3. They provided anonymous code link in Sec. 4.1 is broken. It would have been helpful to see how such a novel and relatively complex algorithm was implemented in practice.
4. Writing/clarity issues: the paper suffers from issues in writing quality, including clarity and typos, discussed next. For example:
    1. The paper can be a bit challenging to read, not because of technical details, but because redundant information is often provided (see for example, "solution overview" and "solution details" could certainly be shortened and combined in 3.2)
    2. Many sentences are terse and relatively telegraphic. The paper is still understandable, but could have better flow and general writing.
    3. There is a too-long run-on sentence in "To improve SDG for rare events, leveraging expert knowledge such as clinical opinion on …, has immense potential" in the introduction.
5. Typos/formatting issues:
    1. In the abstract: "Although extensively studied in image classification, there is a lack of prior work on SDG for rare event or image classification in imbalanced dataset"
        1. also, this is phrased confusingly. maybe say extensively studied in classification of balanced datasets?  But is having an imbalanced dataset not just kind of a triviality?
    2. also in abstract "integrates domain-invariant knowledge on rare event" another typo.
    3. Also in the intro, page 2 "with DL using pre-trained large vision model (LVM) "
    4. There seems to be an issue in how in-text citations are formatted: no spacing between them, not parenthetical, etc. Makes it a bit harder to read the paper.
    5. Why are the equations written with such tiny text? They require zooming in just to read. I'm not sure if this formatting modification is allowed for ICLR.

**Questions:**

1. Is "single domain generalization" a common term when describing your problem scenario? it sounds like it describes generalizing TO a single domain, not training on a single one and being able to generalize to unknown ones.

---

### Official Review · Reviewer_beHZ · 2024-11-03

**Soundness:** 2
**Presentation:** 2
**Contribution:** 2
**Rating:** 3
**Confidence:** 3

**Summary:**

This paper discusses Single Domain Generalization (SDG) in the context of rare event classification, particularly in medical diagnosis, where limited data and imbalanced datasets pose significant challenges. The authors propose a method that utilizes a pre-trained large vision model to rank classes by their similarity to rare events, enabling focused classification. By integrating domain-invariant expert knowledge with data-driven deep learning, the approach improves model robustness and performance, even with scarce data. A case study on detecting seizure onset zones using fMRI data shows that this method achieves an average F1 score of 90.2%.

**Strengths:**

This paper addresses an interesting and important problem: how to leverage large vision models (LVMs) to tackle rare diseases, which involve imbalanced, infrequent cases and domain shifts. While there are many SDG methods available, the authors claim that none have been specifically applied to rare cases in medical data. It appears that the authors are addressing a new problem.

**Weaknesses:**

Problematic Settings

1. While there are currently no methods addressing SDG in the context of limited data and rare diseases, this setting appears to combine elements of SDG with few-shot imbalanced classification. If a SDG method is sufficiently robust, it could potentially be integrated with existing imbalanced learning or rare disease fine-tuning techniques to effectively address this issue. However, the authors lack experiments to validate this point, which raises questions about the relevance of this particular setting in the field. The authors have the opportunity to tackle a significant challenge: how to integrate large vision models (LVMs) for rare disease classification, taking into account that rare diseases can vary in size, may be in-domain or out-of-domain, and may be either imbalanced or balanced, thereby providing a more comprehensive framework.

Limited Discussions and Comparisons of Results

2. The authors have not compared their approach with existing SDG and imbalanced learning methods, which diminishes the convincing nature of their experiments and calls into question their claim of being state-of-the-art (SOTA).

**Questions:**

Please see my comments above

---

### Official Review · Reviewer_iCnz · 2024-11-04

**Soundness:** 2
**Presentation:** 3
**Contribution:** 2
**Rating:** 3
**Confidence:** 3

**Summary:**

This paper presents RareSaGe, a novel approach for SGD in detecting rare events in medical imaging. First, this method handles the most similar classes by using a pre-trained large vision model (LVM) to rank classes based on their similarity to the rare event class, which aims to effectively classify the rare event. Then, it further integrates expert knowledge with deep learning to enhance robustness and address challenges related to limited data. Experiments conducted on multi-center datasets for seizure onset zone (SOZ) detection in fMRI data demonstrate that RareSaGe achieves high generalization performance, significantly outperforming other models.

**Strengths:**

1. The manuscript is written in clear English and is relatively easy to follow.
2. The experimental results demonstrate the effectiveness of the proposed framework.
3. The motivation for developing the method makes sense

**Weaknesses:**

1. The title is focused on MEDICAL IMAGING, but the data is too singular, validated only on fMRI data and limited to two categories (two centers).
2. The introduction to the dataset in Section 4.1 and the Supplementary Material is relatively limited.
2. There are missing many important ablation studies, such as validating the effectiveness of the two types of expert knowledge on the method.

**Questions:**

1. The title is focused on MEDICAL IMAGING, but the data is too singular, validated only on fMRI data and limited to two categories (two centers).
2. In Section 4.1, the link to the dataset is unavailable; it was not previously a publicly available dataset. The introduction to the dataset in Section 4.1 and the Supplementary Material is relatively limited.
3. Additionally, there are no visual results of the dataset presented in the manuscript, leading to insufficient feasibility of the results.
4. In Section 3.4.2 EXPERT KNOWLEDGE ON RS-fMRI, it is mentioned that "These locations can be extracted employing established image processing algorithms." The visual results obtained from these algorithms for specific locations should be presented.
5. There are many important ablation studies missing, such as validating the effectiveness of the two types of expert knowledge on the method.

---

### Official Review · Reviewer_EJaS · 2024-11-04

**Soundness:** 3
**Presentation:** 2
**Contribution:** 2
**Rating:** 3
**Confidence:** 3

**Summary:**

The paper addresses Single Domain Generalization (SDG) for rare event detection in medical imaging, proposing the RareSaGe framework. By combining a large vision model with expert knowledge, the framework handles domain shifts without requiring multi-source data. Applied to seizure onset zone (SOZ) detection in fMRI, RareSaGe achieves good results.

**Strengths:**

The paper effectively integrates pre-trained large vision models with expert domain knowledge to manage imbalanced data and domain variability in a novel way, with potential for practical application.

**Weaknesses:**

1. The method is tested only on fMRI for SOZ detection; broader evaluation on other modalities and medical events would strengthen the generalizability claims.
2. The use of class-wise entropy to manage overlap between rare and similar classes adds complexity that could reduce model transparency. It’s unclear how sensitive the model is to changes in entropy thresholds and whether misclassifications may occur when entropy values between classes are close.
3. Some sections lack clarity in wording; additionally, when citing multiple references, please use “;” to separate different sources.

**Questions:**

1. When the overlap class closely resembles the rare event class, how does the model handle potential misclassifications?
2. How adaptable is the RareSaGe framework for detecting rare events in other medical imaging tasks, such as tumor detection or identifying rare anomalies in radiographs? Would additional model adjustments be needed to support such tasks?

---

### Meta-Review · Area_Chair_5SxH · 2024-12-14

**Metareview:**

Dear authors,

Thank you for submitting the draft.  The reviewers' rankings indicated that the draft is not ready for publication at this stage.

The draft proposes a method for Single Domain Generalization for rare event detection or image classification in imbalanced dataset. All reviewers have raised concerns about the work. eoxC provided a quite detailed review. A repeated concern was methodology being validated only on fMRI (e.g. iCnz,  EJaS). To justify the title "Event Detection in Medical Imaging" a wider set of experiments is needed or authors can reduce the scope of their work to fMRI only. beHZ questioned that without comparison with existing SDG and imbalanced learning methods, can the proposed method be called SOTA. eoxC raised similar concerns. The authors did not provide feedback.

We hope comments by reviewers will help improve the draft.


regards

AC

**Additional Comments On Reviewer Discussion:**

All reviewers agree draft is not ready for publication. Authors did not provided any feedback.

---

### Decision · Program_Chairs · 2025-01-22

Reject